# Changes in inflammatory gene expression in brain tissue adjacent and distant to a viable cyst in a rat model for neurocysticercosis

**Rogger P. Carmen-Orozco**[1,2,3], **Danitza G. Dávila-Villacorta**[1], **Ana D. Delgado-Kamiche**[1], **Rensson H. Celiz**[1], **Grace Trompeter**[4], **Graham Sutherland**[5‡], **Cesar Gavídia**[6], **Hector H. Garcia**[1,7], **Robert H. Gilman**[1,5,8], **Manuela R. Verástegui**[1] *, for the Cysticercosis Working Group in Peru

**1** Infectious Diseases Laboratory Research-LID, Faculty of Science and Philosophy, Alberto Cazorla Talleri, Universidad Peruana Cayetano Heredia, Lima, Perú, **2** Cellular and Molecular Medicine Program, Johns Hopkins University School of Medicine, Baltimore, Maryland, United States of America, **3** Solomon H. Snyder Department of Neuroscience, Johns Hopkins University School of Medicine, Baltimore, Maryland, United States of America, **4** Jacobs School of Medicine and Biomedical Sciences, University at Buffalo, Buffalo, New York, United States of America, **5** Department of International Health, Bloomberg School of Public Health, The Johns Hopkins University, Baltimore, Maryland, United States of America, **6** School of Veterinary Medicine, Universidad Nacional Mayor de San Marcos, Lima, Perú, **7** Cysticercosis Unit, Instituto Nacional de Ciencias Neurologicas, Lima, Perú, **8** Asociación Benéfica PRISMA, Lima, Perú

‡ Unavailable.
* manuela.verastegui@upch.pe

## Abstract

### Background

The parasite *Taenia solium* causes neurocysticercosis (NCC) in humans and is a common cause of adult-onset epilepsy in the developing world. Hippocampal atrophy, which occurs far from the cyst, is an emerging new complication of NCC. Evaluation of molecular pathways in brain regions close to and distant from the cyst could offer insight into this pathology.

### Methods

Rats were inoculated intracranially with *T. solium* oncospheres. After 4 months, RNA was extracted from brain tissue samples in rats with NCC and uninfected controls, and cDNA was generated. Expression of 38 genes related to different molecular pathways involved in the inflammatory response and healing was assessed by RT-PCR array.

### Results

Inflammatory cytokines IFN-γ, TNF-α, and IL-1, together with TGF-β and ARG-1, were over-expressed in tissue close to the parasite compared to non-infected tissue. Genes for IL-1A, CSF-1, FN-1, COL-3A1, and MMP-2 were overexpressed in contralateral tissue compared to non-infected tissue.

**Data Availability Statement:** All relevant data are within the manuscript and its Supporting Information files.

**Funding:** This work was supported by INNOVATE PERU grant number: N-135 PNICP-PIAP 2015, MRV; CIENCIACTIVA PERU grant number: 118-2015-FONDECYT, MR V, the National Institutes of Health grants number: 5D43 TW006581, RH G and, U19AI129909 (Peru-JHU TMRC Program), HH G. The funders had no role in study design, data collection and analysis, decision to publish, or preparation of the manuscript.

**Competing interests:** The authors have declared that no competing interests exist. Author Graham Sutherland was unable to confirm their authorship contributions. On their behalf, the corresponding author has reported their contributions to the best of their knowledge.

## Conclusions

The viable cysticerci in the rat model for NCC is characterized by increased expression of genes associated with a proinflammatory response and fibrosis-related proteins, which may mediate the chronic state of infection. These pathways appear to influence regions far from the cyst, which may explain the emerging association between NCC and hippocampal atrophy.

## Author summary

*Taenia solium* is a parasite that can infect human brain causing neurocysticercosis. Neuro-cysticercosis is a common cause of adult-onset epilepsy in the developing world. This infection elicits several cellular and molecular changes as a result of inflammation or para-site-host interaction, which can cause clinical symptoms, such as seizures. Most of these changes have been found in the tissue surrounding to the cysticercus, however, some pathologies, like hippocampal atrophy, which occurs in parts of the brain far from the cyst, are emerging as new complication in NCC patients. Using a rat model, the authors assessed the expression of genes related to different molecular pathways involved in the inflammatory response and healing by the RT-PCR array technique. They found increased expression of genes associated with inflammation and scar tissue formation in tissue surrounding the cyst, as well as tissue far from the cyst, when compared to non-infected brain tissue. This study provides new insights into the inflammatory changes that occur in brain tissue far from viable cysts and may provide evidence for the emerging association between NCC and hippocampal atrophy.

## Introduction

Neurocysticercosis (NCC) is caused by *Taenia solium*, a parasite endemic to Latin America, Africa, and regions of Asia [1,2]. Although NCC is both preventable and eradicable, world-wide, it is estimated that as many as 3 million people currently suffer from seizures caused by this disease [3]. It is thought to be responsible for up to one third of all epilepsy cases in these regions, and is the most common risk factor for adulthood-onset epilepsy [4].

The life cycle of *T. solium* begins when humans ingest parasitic cysts that are present in undercooked, infected pork. The cysts mature into adult tapeworms after attaching themselves to the mucosa of the upper small intestine, where they begin to grow and develop proglottids, each of which may contain up to 50,000 ova that are continuously expelled in the host excre-ment [1,5]. Although the natural intermediate hosts of *T. solium* are pigs, humans may inad-vertently take their place by ingesting ova. Once inside the intestinal tract, the ova transform into oncospheres and upon passing through the intestinal wall are carried throughout the bloodstream to different parts of the body, including the central nervous system, where in just a few months they form viable cysts capable of surviving for many years [1,5]. In the brain, cysts are most commonly found in the parenchyma, the region receiving the largest blood sup-ply [1].

In NCC patients, viable cysts do not appear to be associated with inflammation, and thus ostensibly do not cause clinical manifestations in their host [6]. Over time, and for reasons yet to be determined, the immune system eventually recognizes the cysts, producing the cysts

degeneration accompanied by an associated inflammation[6]. Different imaging techniques such as magnetic resonance imaging (MRI) and computerized tomography (CT) permit researchers to evaluate the degree of inflammation and cyst degeneration, but these methods give no clue as to the actual underlying biochemical mechanisms that occur as the disease progresses [7–9]. Therefore, the study of viable cysts and the various molecular pathways stimulated in their presence can provide pivotal information about NCC pathophysiology and its symptoms.

Our group has shown in the rat model that changes such as blood-brain-barrier disruption, angiogenic factor overexpression, and axonal swellings occur in the presence of viable cysts [10,11]. Most of these changes have been found and reported in the tissue surrounding the cyst. However, hippocampal atrophy occurring far from the cyst is emerging as new complication in NCC patients. [12–14]. We set out to evaluate the different molecular pathways in brain regions distant from the cyst in order to better understand the pathogenesis of this outcome.

The following work examined gene expression by a real time PCR array of 38 genes related to different molecular pathways involving cytokine production, fibrosis, angiogenesis, oxidative stress metabolism and apoptosis in the tissue surrounding viable cysts and its contralateral region in a rat model for NCC. Uninfected rats were used as a control for basal gene expression. Tissue samples contralateral to the cysts were examined in order to test the hypothesis that cysticerci can exert influence in gene expression even in regions far from the parasite.

## Materials and methods

### Ethics statement

Animal procedures were conducted according to Universidad Peruana Heredia ethics procedures (IATA number 64637). The protocol was approved by the IACUC of the Universidad Peruana Cayetano Heredia in Lima, Peru.

### Animal infection

Nineteen animals from 12 to 15 days old were obtained from the UPCH animal care facility and were inoculated with 120 *T. solium* activated onchospheres following a previously reported protocol (9). Briefly, enveloped *T. solium* eggs were treated with sodium hypochlorite 0.75% solution, and then activated with porcine bilis and pancreatine solution. Oncospheres were injected intracraneally in the bregma region following administration of anesthesia (ketamine 150 mg/kg and xylazine 20 mg/kg).

### Tissue collection

Four months after infection, rats were euthanized using ketamine and xylazine, perfused with PBS and the brain was removed inside a laminar flow cabinet. For non-infected rats (n = 6), all brains neocortices were dissected and small parts of frontal cortex (anterior) and occipital temporal cortex (posterior) were collected in order to have a random sample of control tissue (12 non-infected tissues). For infected rats (n = 13), the tissue surrounding the cyst was dissected and collected and other samples were taken from the contralateral side of the brain, distant from any cyst (n = 13, Fig 1). Only brains carrying 1–2 parenchymal cysts were used. Characteristics of tissue samples taken from each rat are shown in S1 Table. Collected tissues were immersed in RNAlater at 4˚C overnight, and the following day RNAlater was removed and samples were stored at -70˚C, according to manufacturer's instructions (Ambion).

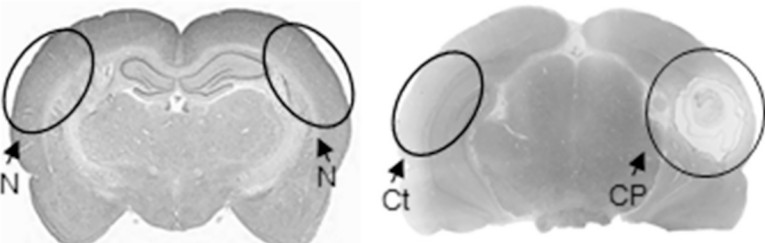

**Fig 1. Tissue collected in this study.** Non-infected tissue (N, n = 12), tissue close to the parasite (CP, n = 13) and, its contralateral tissue (Ct, n = 13).

## RNA extraction and qRT-PCR array

RNA was extracted using a column-based method (RNeasy Microarray Tissue, Qiagen). Briefly, 100mg of tissue was homogenized in 1ml QIAzol lisys reagent, chloroform was added, centrifuged and the aqueous phase was collected and placed in a new tube containing 70% ethanol. The solution was transferred to a spin column, centrifuged and washed two times in kit buffers. RNase-free water was added, and RNA concentration (260nm) and quality (ratio 260/280nm) were measured using NanoDrop. Next, cDNA was generated from 5ug of RNA for each sample using RT2 First Strand Kit (Qiagen), which contained a genomic DNA elimination step. All cDNA and RNA obtained were tested using RT2-RNA-QC-PCR-Array kit (Qiagen, Cat# PARN-999ZC-1), which contained 8 controls used to assess the quality of each sample. Later, cDNA first strand samples that passed quality control evaluation were mixed with 2 × RT2 SYBR Green qPCR Master Mix and RNase-free water. The StepOneplus instrument was used for real time PCR under the following cycling conditions; 10 min at 95˚C, 40 cycles at 95˚C for 15 sec and finally 1min at 60˚C. In total, 38 genes were evaluated for each of the 38 samples, and 5 additional housekeeping genes. Twenty plates of 96 wells were run and the threshold used for Ct calculations was set as 10 times the standard deviation of the baseline (first 10 cycles). NormFinder was used for Housekeeping gene selection and finally, Expression levels for all markers were normalized using beta actin gene, which proved to be the most stable housekeeping gene compared to B2M, HPRT1, LDHA, and RPLP1. Finally, $2^{-\Delta\Delta CT}$ method was chosen for relative gene expression [15].

## Statistics

Comparison between non-infected tissue, tissue close to the parasite, and its contralateral side was performed using Kruskall Wallis, and Benjamini-Hochberg adjustment was used for pairwise comparison. Wilcoxon matched-pairs signed-ranks test was used for comparison between the anterior and posterior areas from non-infected animals and for comparison between the tissue close to the parasite and its contralateral side. QCanvas v1.21 was used for heat map graphing, relative fold of change was converted to a logarithmic scale, and for clustering, average method and spearman rank correlation was chosen.

## Results

### Inflammatory cytokines are overexpressed in NCC model

Notably, inflammatory cytokines such as IFN-γ, TNF-α, colony stimulator factors (CSF-1 and CSF-2), IL-1β, IL-1α, as well as IL-6, which is considered an immunomodulatory cytokine, were overexpressed in the tissue close to the parasite compared to the non-infected tissue (P = 0.037, <0.001, <0.001, 0.006, <0.001, <0.001 and <0.001, respectively). As expected,

these cytokine expression levels were significantly increased in the infected tissue surrounding the cyst when compared to its contralateral side (P = 0.019, 0.003, 0.014, 0.010, 0.018 and 0.004 respectively), except for CSF-2, (P = 0.076). However, for anti-inflammatory cytokines like IL-4 and IL-10; no significant differences in expression were found in the tissue surrounding the parasite compared to its contralateral side and the non-infected tissue (Fig 2).

Among the nitric oxide synthases evaluated; neuronal (nNOS), inducible (iNOS) and endothelial (eNOS), only iNOS was overexpressed in the tissue close to the cyst compared to the non-infected tissue (P<0.001). Furthermore, its competitor enzyme, arginase-1 (ARG-1), was also overexpressed close to the cyst (P = 0.001). No changes were found in other oxidative stress related molecules such as superoxide dismutase 1 (SOD-1) and hypoxia-inducible factor 1-alpha (HIF-1α). Additionally, caspase 7 (CASP-7) gene expression, a protease involved in inflammation and apoptosis, was elevated in the tissue close to the cyst (P = 0.003, Fig 2).

### Fibrosis and angiogenic related genes response

Transforming growth factor beta 1 (TGF-β1), a fibrosis and immunomodulation related gene and its receptor (TGF-βR1) were overexpressed in the tissue close to the parasite compared with non-infected animals (P<0.001, P<0.001 respectively) and with its contralateral side (P = 0.009, 0.018). TGF-β3 was also overexpressed in the tissue close to the parasite compared to its contralateral side (P = 0.012) but no changes compared to the non-infected tissue were found. Additionally, no changes for TGF-β2 were found. As expected, extracellular matrix related genes including fibronectin 1 (FN-1), collagen 1a1 (COL-1A1) and collagen 3a1 (COL-3A1) were increased in comparison to the non-infected tissue (P<0.001, <0.001, <0.001, respectively) and its contralateral side (P = 0.011, 0.006, 0.022, respectively). Evaluation of proteins related to extracellular matrix remodeling showed that matrix metalloproteinase 2 (MMP-2), but not MMP-9, was overexpressed in the tissue close the cyst compared to the non-infected tissue (P<0.001) and to its contralateral side (P = 0.014), (Fig 3).

Angiogenic gene expression showed down regulation of VEGF-A in the tissue close to the parasite compared to the non-infected tissue (P = 0.009); but no changes were found in its receptors (VEGFR-1 and 2), and in the other VEGF genes (VEGF-B and VEGF-C). In addition, angiopoietin-1 (ANGPT-1) and its receptor as well as fibroblast growth factor 2 (FGF-2) showed no changes in their expression. However, expression of the angiopoietin-2 gene (ANGPT-2) was increased in the tissue close to the parasite compared to the non-infected tissue and its contralateral side (P = 0.003 and 0,046, respectively).

### Changes in gene expression in the contralateral side of the infected brain

Most of the 38 genes evaluated in the tissue contralateral to the cyst did not show a significant difference in expression compared to the non-infected tissue except for IL-1α, CSF-1, FN1, COL3A1, and MMP-2, which were overexpressed (P = 0.043, 0.038, 0.017, 0.045, 0.040, respectively), and neuropilin-1, which was down regulated (P = 0.019, respectively). To better illustrate the effects on the contralateral tissue, we show a heatmap where each infected tissue is next to its contralateral side (Fig 4) Interestingly, many contralateral tissues presented similar gene expression patterns as its infected tissue close to the parasite (Fig 4, top), and as expected, the rest of the samples showed a similar profile as the non-infected rats (Fig 4, middle). A heat map representing all evaluated genes is presented in S1 Fig.

### Discussion

Our study shows, first, that the tissue surrounding the *T. solium* cysticercus exhibits a proinflammatory and fibrosis-related gene profile; and second, that the tissue contralateral to the

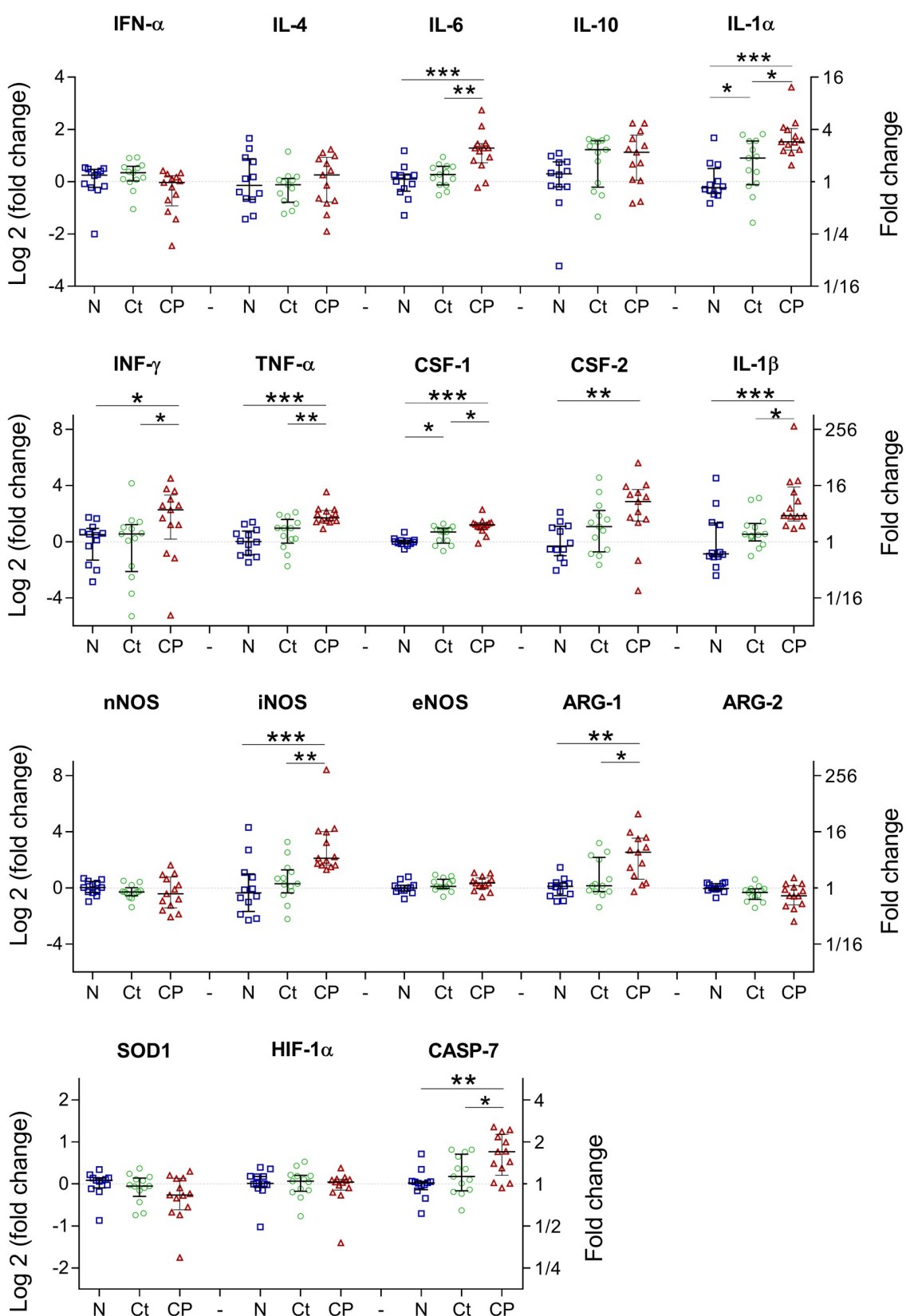

**Fig 2. Inflammatory cytokines and oxidative stress related molecules in NCC.** Pro-inflammatory cytokines were overexpressed in the tissue close to the parasite (CP), effects on contralateral region (Ct) were evaluated and compared against non-infected tissue (N). n = 13 infected tissues for CP and Ct, n = 12 non-infected tissues. *, P<0.05; ** P<0.01; *** P<0.001.

cyst also undergoes differential expression of the gene profile, suggesting that *T. solium* cysticercus can exert pathological changes not only in its vicinity but also in distant brain regions such as the contralateral hemisphere. It is possible that various host-parasite interactions cause pathological changes even without producing symptoms in the patient. Since previous studies have only focused on processes that occur in non-viable, degenerating cysts, we report here a descriptive study that evaluated the changes in different molecular pathways due to the presence of viable parenchymal cysts.

We observed that most of the pro-inflammatory-associated cytokines examined (IFN-γ, TNF-α, CSF-1, CSF-2, IL-1β, IL-1α) were increased in the area surrounding the viable cyst when compared to the non-infected tissue. It has long been thought that viable cysts, before they degenerate, exhibit little or no inflammation in the surrounding tissue [6,16,17], however, our results challenge this notion. Perhaps instead, these cytokine reactions are part of the immunoregulatory mechanisms that allow for a permissive environment in the asymptomatic host [18,19]. It is also plausible that the cyst itself secretes molecules that promote localized immunosuppression or host tolerance. At this stage of infection (i.e. the viable cyst stage) the $T_H1$ response appears to represent a host inflammatory reaction that while quantifiable at the molecular level, remains subclinical and ineffective in destroying the parasite. Another explanation for this observation could be that the cyst is already starting to degenerate even though we saw no clinical evidence of this in the rat. Finally, a third explanation could be that while the rat is the best mammalian model we have at our disposal to study this complex disease process, it is an atypical host for *T. solium*, and the immune response triggered in the rat may be different than that of humans.

Interestingly, no differences were found in expression of the $T_H2$ associated cytokine gene IL-4, even though a predominantly $T_H2$ response is more often associated with asymptomatic NCC [20,21]. The $T_H2$ response is important in protecting against extracellular helminthic parasites through suppression of the inflammatory $T_H1$ response, neutralization of toxins, and defense of the host against damage [22,23]. To further understand the role of the $T_H2$ cytokines in the pathogenesis of the viable cyst, future studies should evaluate all cytokines in this group, including IL-5 and IL-13.

TGF-β, along with IL-10 play important roles in the T regulatory ($T_{reg}$) pathways of the inflammatory response. Here, we showed increased TGF-β1 and its receptor, TGF-βR1 gene expression in tissue surrounding the cysts compared to contralateral tissues and controls. No significant differences in IL-10 gene expression were observed between infected and control tissue. TGF-β genes are implicated in a variety of CNS pathologies, including epilepsy and neuroinflammation [24–26], but have also been shown to promote neuroprotection and brain maintenance [27]. Bar-Klein, et al. demonstrated the contributory role of albumin-mediated TGF-β1 signaling in epileptogenesis following vascular insult [24]. On the other hand, the potential neuroprotective effects of TGF-β, through regulation of apoptotic protein expression, have been demonstrated in the setting of cerebral ischemia [28]. However, other studies have shown that both parasite and host-secreted TGF-β play important roles in parasite infection. Proteins secreted by parasites in the host tissue can promote $T_{reg}$ induction through the TGF-β pathway. Human TGF-β has been shown to promote survival of *T. solium* cysticerci *in vitro* [29]. Additional investigation is necessary to fully elucidate the complex role of TGF-β in NCC pathogenesis.

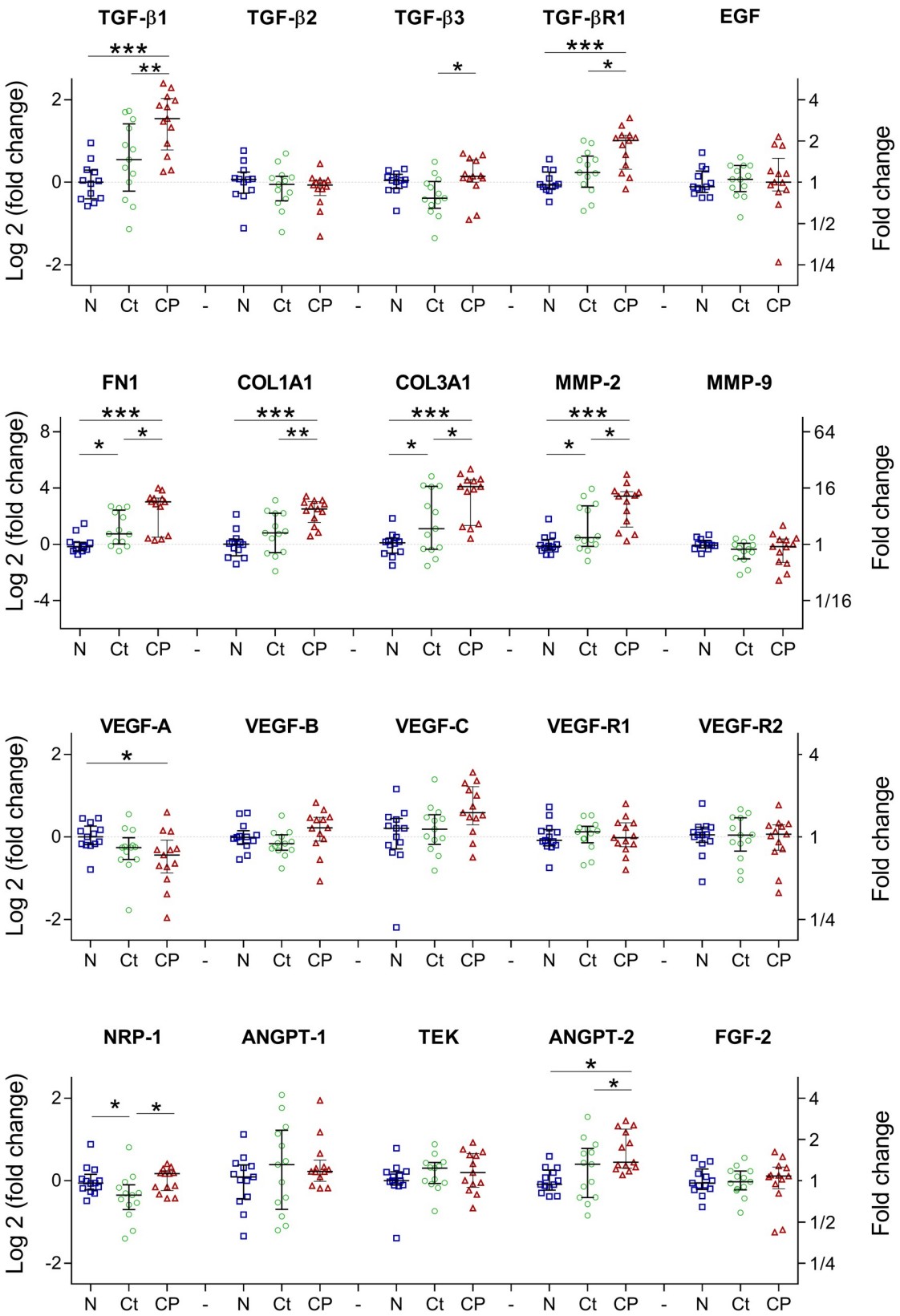

**Fig 3. TGF-β and fibrosis-related genes were overexpressed in NCC.** Tissue close to the parasite (CP) and contralateral region (Ct) were compared against non-infected tissue (N). n = 13 infected tissues for CP and Ct, n = 12 non-infected tissues. *P<0.05; ** P<0.01; *** P<0.001).

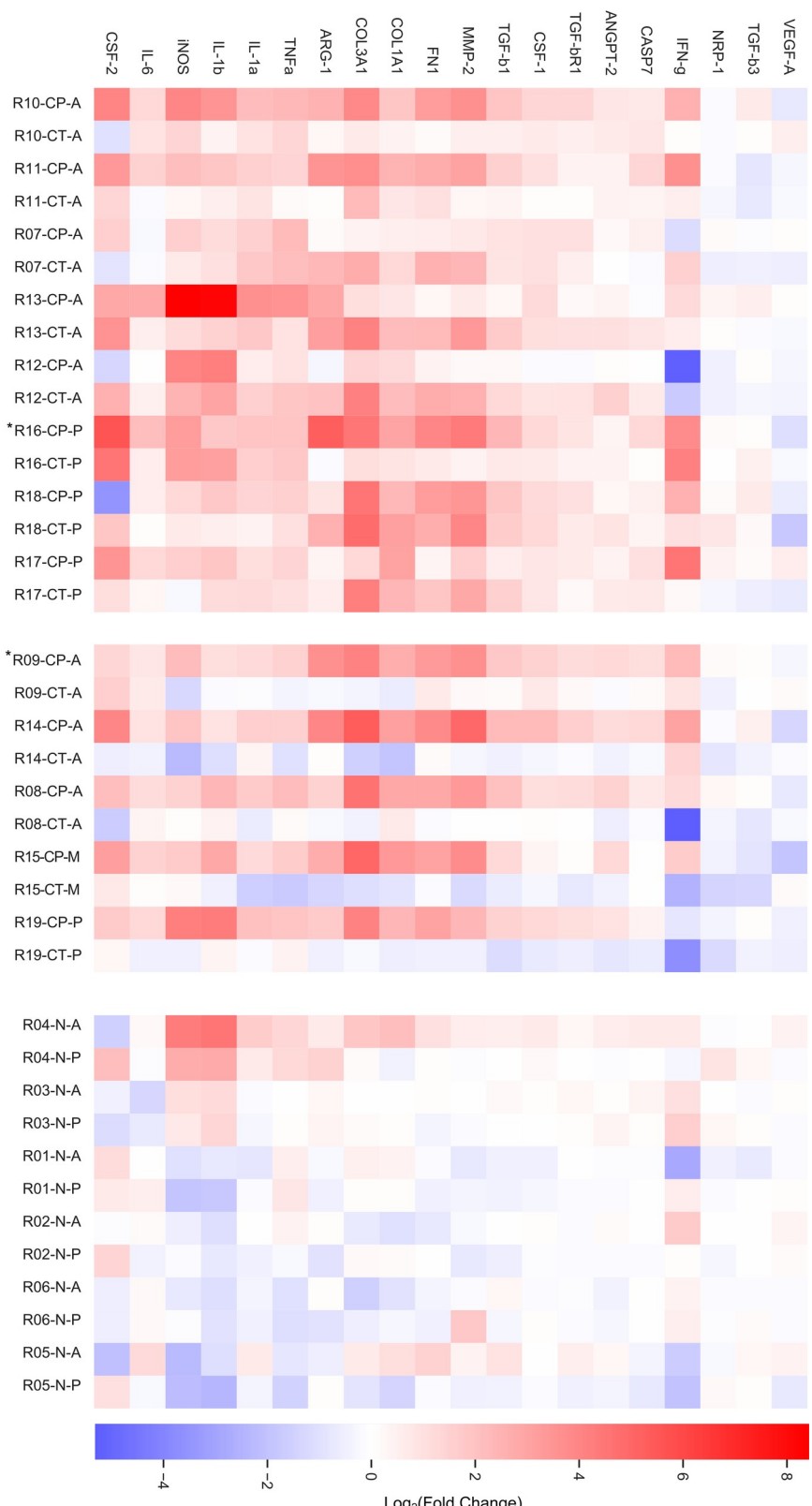

**Fig 4. NCC rat model shows a predominance of pro-inflammatory and fibrosis-related response with changes in gene expression tissue distant to the parasite.** Genes with significant changes in gene expression compared to non-

infected tissue or to the contralateral side are displayed. Each tissue close to the parasite (CP) is represented next to its contralateral tissue (CT). The upper part of the heatmap shows tissue where the contralateral presents an inflammatory and fibrosis related response. The middle of the graph shows samples where the contralateral tissue seems to not be affected by the presence of the parasite. At the bottom not-infected tissue is shown (N). Three different regions of the brain were used: frontal cortex (anterior, A), parietal temporal cortex (medial, M) and occipital temporal (posterior, P). *, indicates rats with two parenchymal cysts. Bar represents log2 of fold change.

Regarding NOS gene expression, our results support the idea that a pro-inflammatory pathway is governing the viable NCC cyst response, as iNOS was overexpressed. In neurodegenerative disorders such as Alzheimer's disease, iNOS stimulates overproduction of NO, yet neuroprotective mechanisms of NO through a number of molecular pathways have also been demonstrated [30]. In this study, ARG-1, a known enzyme competitor of iNOS, was also overexpressed. Madan reported that arginase I levels are elevated after a brain injury (TBI), and ARG-2 maintains its basal levels. Quierie, et al. reported similar findings in the setting of stroke [31,32]. Arginase gene expression in this study followed the same pattern. Furthermore, high levels of ARG-1 following brain insult indicate the presence of M2 type macrophages, known to play an important role in tissue remodeling and suppression of the inflammatory response [31,32]. Thus, arginase 1 may be participating in both the anti-inflammatory response and fibrogenesis pathways, which is perhaps the main role this protein plays here [30,31].

Brain tissue in the presence of NCC stimulates formation of a fibrosis layer, which acts as an interface between the parasite and host tissue. As expected, fibrosis-related genes such as COL-1A1, COL-3A1, and FN-1 were overexpressed surrounding the cyst. Among the VEGF genes, only VEGF-A showed a difference in expression between tissue surrounding the cyst and normal tissue. Surprisingly, gene expression of VEGF-A was slightly down regulated. This contrasts a previous report by our group which showed angiogenesis in NCC-infected brain slices as well as an increase in VEGF-A protein expression surrounding the cyst [10]. Even though we did not find significantly increased levels of gene expression associated with angiogenesis in this study, it is evident that in the tissue surrounding the cyst, angiogenesis is present and expected [10,17]. Furthermore, we did not observe a parallel relationship between VEGF-A RNA levels in this study and VEGF-A protein levels reported in our previous study. Thus, it is possible that changes in expression of VEGF-A involve regulatory changes in its RNA half-life or different ratios of protein translation related to the complex expression of VEGF-A gene and its isoforms [33,34].

We have seen a predominant inflammatory response presented as overexpression of IL-1β, IL-1α, IFN-γ, and CSF-1, 2. However, we also observed two important molecules involved in the regulation of the immune system. First TGF-β1, a cytokine presented in regulatory lymphocytes. Second ARG-1, which plays a role in M2 macrophage polarization and is a competitor enzyme of iNOS [35]. Both TGF-β and ARG-1, besides their important roles in regulating the immune response, are known to play a role in fibrosis formation [36,37]. Thus, TGF-β1 and ARG-1 could play key roles in directing the anti-inflammatory profile and chronic features presented in viable NCC.

Of the 38 genes evaluated, we found five–IL-1a, CSF-1, FN-1, COL-3A1, and MMP-2 –that were overexpressed in the contralateral tissue when compared to gene expression in non-infected tissue. These results revealed that brain tissue distant to viable *T. solium* cysts can be compromised and may help explain the possible mechanism of hippocampal atrophy found in NCC patients, chronic seizures are also thought to lead directly to hippocampal atrophy [38,39]. There is still debate whether hippocampal atrophy found in NCC cases is the result of an inflammatory process, changes in brain activity, or if it is due to the presence of the parasite

itself. Furthermore, different studies support the notion that the cysticercus can stimulate brain responses at farther regions [40,41]. We recently reported on findings that show axonal swelling in the presence of NCC extends beyond regions showing inflammatory changes, suggesting there may be contribution by anterograde and retrograde transport in promoting hippocampal atrophy [11]. Additionally, molecular mechanisms known to promote hippocampal atrophy include inflammation and production of collagen proteins. We found overexpression of COL-3A1, which in hippocampal tissues has the ability to regulate network structure [42] and whose overexpression can result in neuronal loss [43]. This is the first report we are aware of that showed that changes in gene expression of inflammatory cytokines occur in brain tissue distant to the cyst, adding to the body of evidence supporting NCC's role in hippocampal atrophy.

This study investigated the differential expression of 38 genes related to different molecular pathways in the setting of NCC. However, in order to understand the complexity of the pathways involved in NCC pathogenesis, further investigation with robust techniques such as high-throughput screening like RNAseq or proteomics is needed [44,45]. Here, we demonstrated that regions distant to the parasite, in this case the contralateral side of the brain, are affected and induce changes in gene expression. One limitation of this approach is that while gene expression is known to vary between brain hemispheres [46–48], we did not compare the gene profile between the left and right hemispheres. This decision was made due to the understanding that in NCC, cysts appear to be randomly oriented and predilection to one specific side has not been observed.

Finally, we can conclude that the viable cysticerci stage of NCC in the rat model is characterized by increased expression of genes associated with a proinflammatory response and fibrosis-related proteins, including collagen, TGF-β, and ARG-1, which together may mediate the chronic state of infection. These pathways appear to influence regions far from the cyst, providing a possible explanation for the emerging association between NCC and hippocampal atrophy.

## Supporting information

**S1 Table. Number of tissues used in this study.** Tissue close to the parasite (CP), contralateral tissue (CT) and non-infected tissue were collected (N) from different brain areas anterior (A), medium (M) and posterior (P). # *Cysts* refers to the number of cysts found in each rat and *P cysts* refers to the number of cysts located in the parenchymal tissue.
(DOCX)

**S1 Fig. Heatmap represents all genes evaluated in this study.** Gene responses in tissue close to the parasite (CP) form a cluster but also shared its expression with its contralateral tissue (CT). Some CT share the gene profile presented in non-infected tissue (N). Three different regions of the brain were used anterior (A), medial (M) and posterior (P). Bar represents log2 of fold change.
(TIF)

## Acknowledgments

We would like to thank Cesar Quispe Asto, Gino Castillo Vilca, Monica Criollo Joaquin and Lizbeth Fustamante Fernandez as part of the Cysticercosis Working Group in Peru.

## Author Contributions

**Conceptualization:** Rogger P. Carmen-Orozco, Robert H. Gilman, Manuela R. Verástegui.

**Data curation:** Rogger P. Carmen-Orozco, Danitza G. Dávila-Villacorta.

**Formal analysis:** Rogger P. Carmen-Orozco.

**Funding acquisition:** Cesar Gavídia, Hector H. Garcia, Robert H. Gilman, Manuela R. Verástegui.

**Investigation:** Rogger P. Carmen-Orozco, Danitza G. Dávila-Villacorta, Ana D. Delgado-Kamiche, Rensson H. Celiz.

**Methodology:** Rogger P. Carmen-Orozco, Danitza G. Dávila-Villacorta.

**Project administration:** Robert H. Gilman, Manuela R. Verástegui.

**Resources:** Cesar Gavídia, Hector H. Garcia, Robert H. Gilman, Manuela R. Verástegui.

**Software:** Rogger P. Carmen-Orozco.

**Supervision:** Robert H. Gilman, Manuela R. Verástegui.

**Validation:** Rogger P. Carmen-Orozco.

**Visualization:** Rogger P. Carmen-Orozco.

**Writing – original draft:** Rogger P. Carmen-Orozco, Danitza G. Dávila-Villacorta, Ana D. Delgado-Kamiche, Rensson H. Celiz, Grace Trompeter, Graham Sutherland.

**Writing – review & editing:** Rogger P. Carmen-Orozco, Danitza G. Dávila-Villacorta, Ana D. Delgado-Kamiche, Rensson H. Celiz, Grace Trompeter, Graham Sutherland, Cesar Gavídia, Hector H. Garcia, Robert H. Gilman, Manuela R. Verástegui.

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
