## [Decision Letter · Decision Letter 0]

9 Jan 2020

Dear PhD Verastegui:

Thank you very much for submitting your manuscript "Changes in inflammatory gene expression in brain tissue adjacent and distant to a viable cyst in a rat model for neurocysticercosis" (PNTD-D-19-01926) for review by PLOS Neglected Tropical Diseases. Your manuscript was fully evaluated at the editorial level and by independent peer reviewers. The reviewers appreciated the attention to an important topic but identified some aspects of the manuscript that should be improved.

We therefore ask you to modify the manuscript according to the review recommendations before we can consider your manuscript for acceptance. Your revisions should address the specific points made by each reviewer.

(1) A letter containing a detailed list of your responses to the review comments and a description of the changes you have made in the manuscript.

(2) Two versions of the manuscript: one with either highlights or tracked changes denoting where the text has been changed (uploaded as a "Revised Article with Changes Highlighted" file ); the other a clean version (uploaded as the article file).

(3) If available, a striking still image (a new image if one is available or an existing one from within your manuscript). If your manuscript is accepted for publication, this image may be featured on our website. Images should ideally be high resolution, eye-catching, single panel images; where one is available, please use 'add file' at the time of resubmission and select 'striking image' as the file type. 

Please provide a short caption, including credits, uploaded as a separate "Other" file. If your image is from someone other than yourself, please ensure that the artist has read and agreed to the terms and conditions of the Creative Commons Attribution License at http://journals.plos.org/plosntds/s/content-license (NOTE: we cannot publish copyrighted images). 

(4) Appropriate Figure Files 

Please remove all name and figure # text from your figure files upon submitting your revision. Please also take this time to check that your figures are of high resolution, which will improve both the editorial review process and help expedite your manuscript's publication should it be accepted. Please note that figures must have been originally created at 300dpi or higher. Do not manually increase the resolution of your files. For instructions on how to properly obtain high quality images, please review our Figure Guidelines, with examples at: http://journals.plos.org/plosntds/s/figures

While revising your submission, please upload your figure files to the Preflight Analysis and Conversion Engine (PACE) digital diagnostic tool, https://pacev2.apexcovantage.com/ PACE helps ensure that figures meet PLOS requirements. To use PACE, you must first register as a user. Then, login and navigate to the UPLOAD tab, where you will find detailed instructions on how to use the tool. If you encounter any issues or have any questions when using PACE, please email us at figures@plos.org.

We hope to receive your revised manuscript by Mar 09 2020 11:59PM. If you anticipate any delay in its return, we ask that you let us know the expected resubmission date by replying to this email.

To submit your revised files, please log in to https://www.editorialmanager.com/pntd/

Sincerely,

Adly M.M. Abd-Alla, Prof asso.

Guest Editor

Mar Siles-Lucas

Deputy Editor

Reviewer's Responses to Questions

**Key Review Criteria Required for Acceptance?**

**Methods**

-Are the objectives of the study clearly articulated with a clear testable hypothesis stated?

-Is the study design appropriate to address the stated objectives?

-Is the population clearly described and appropriate for the hypothesis being tested?

-Is the sample size sufficient to ensure adequate power to address the hypothesis being tested?

-Were correct statistical analysis used to support conclusions?

-Are there concerns about ethical or regulatory requirements being met?

Reviewer #1: The objectives are well articulated with a clear testable hypothesis. 

Study design appropriate.

Sample size is small, but the results striking and is an animal model. 

Statistical anaylsis appropriate

Reviewer #2: I have no concerns regarding the methods of the study.

**Results**

-Does the analysis presented match the analysis plan?

-Are the results clearly and completely presented?

-Are the figures (Tables, Images) of sufficient quality for clarity?

Reviewer #1: Analysis plan is appropriate.The results are clearly presented. 

Figures are sufficient.

Page 6 - the two lines 141-142 give the results and use symbols (*,#,etc) - I found this hard to follow. 

Page 7- 159-161 - same comments applies to page 6.

Reviewer #2: I have no concerns with the results section.

**Conclusions**

-Are the conclusions supported by the data presented?

-Are the limitations of analysis clearly described?

-Do the authors discuss how these data can be helpful to advance our understanding of the topic under study?

-Is public health relevance addressed?

Reviewer #1: The conclusions supported by the data supported and has important implications for the field of NCC. Hippocampal atrophy is an emerging field and more data is needed. The authors do a nice job of laying this out for the reader. The discussion of the results, which are very interesting, can help us understand pathogenesis hippocampal atrophy in the setting of NCC and the public health relevance is addressed. 

The limitations is are addressed.

Reviewer #2: No concerns with the conclusions section.

**Editorial and Data Presentation Modifications?**

Reviewer #1: Page 4 - line 73 - authors note that leading to a cascade of inflammatory responses that cause cyst degeneration - I pulled reference 6 and there is no discussion of inflammatory response.

Reviewer #2: Minor criticisms are as follows:

1. The sentence on lines 79-81 has a lot of commas and is confusing to read. I would separate into 2 sentences.

2. Line 231 – would add the phrase “in CT samples” to the end of the sentence, “As expected, fibrosis-related genes…..”

3. Line 246 – would change “NCC” to say “viable NCC” as the expected inflammatory profile would differ significantly in degenerating or calcified lesions.

4. Line 250-251, would add that chronic seizures are also thought to lead directly to hippocampal atrophy

5. Lines 257-259 – between “found” and “changes” would put the word that. 

Or could change to “this is the first report… that showed that changes”

6. Line 261 – “in order understand” should be changed to “in order to understand”

**Summary and General Comments**

Reviewer #1: The work presented is interesting. The hypothesis is based on clinical observations on the association of hippocampal atrophy in the setting of NCC. The authors set up a lovely experiment to begin to unravel pathogenesis. The results support that there is increased expression of inflammatory cytokines is over expressed in tissue close to the parasite - which is expected. But, genes for IL-1a, CSF-1, FN-1, COL-3A1, and MMP-2 were over expressed in contraleral tissue compared to non-infected tissue. This is quite novel and the results have clinical implications.

Reviewer #2: The paper by Carmen-Orozco et al discusses an extremely important topic, that of the role of inflammation in the pathophysiology of NCC. The authors are the first to demonstrate an upregulation of pro-inflammatory and pro-fibrosis genes on the contra-lateral side to a viable NCC lesion. This provides extremely important information regarding the possible mechanisms by which epilepsy can develop in NCC. Overall, the paper is well written and provides significant new information that will be very useful to those studying NCC and/or epilepsy.

I have a few chief questions for the authors:

1. Did any of the infected rats experience seizures during the study? Although seizures are more common with degenerating and calcified lesions, they can still occur with viable lesions and this could be an important confounder as seizures are also known to trigger inflammation.

2. Line 77 – stated that the study of viable lesions can “provide pivotal information about NCC etiology and its symptoms.” I would change etiology to pathophysiology, as we already know the etiology of NCC.

3. Methods section, line 100 – They state that for all infected rats, “the tissue surrounding the cyst was dissected and collected.” However, some of the rats had 2 cysts – for these rats, did they collect both cysts or just one? And if they collected only one, how did they chose between cysts? Also, did they do any analyses to compare those with 1 cyst to those with 2? Was there greater inflammation if there were 2 cysts?

4. Discussion, lines 235-237 – The authors state that, even though they did not find increased gene expression of angiogenesis, “it is evident that in the tissue surrounding the cyst, angiogenesis is present and expected.” How was this evident that angiogenesis was present? Was this seen on pathology? 

5. Figure 4 – it is impossible, looking at the figure the way it is, to compare the groups. Would separate the CP, CT, and N groups into three separate parts of the figure so you can actually compare between the groups.

PLOS authors have the option to publish the peer review history of their article (what does this mean?). If published, this will include your full peer review and any attached files.

Reviewer #1: Yes: Christina coyle

Reviewer #2: No

---

## [Editor Report · Decision Letter 1]

7 Mar 2021

Dear PhD Verastegui,

We are pleased to inform you that your manuscript 'Changes in inflammatory gene expression in brain tissue adjacent and distant to a viable cyst in a rat model for neurocysticercosis' has been provisionally accepted for publication in PLOS Neglected Tropical Diseases.

Best regards,

Adly M.M. Abd-Alla, Prof asso.

Associate Editor

Mar Siles-Lucas

Deputy Editor

---

## [Editor Report · Acceptance letter]

22 Apr 2021

Dear PhD Verastegui,

We are delighted to inform you that your manuscript, "Changes in inflammatory gene expression in brain tissue adjacent and distant to a viable cyst in a rat model for neurocysticercosis," has been formally accepted for publication in PLOS Neglected Tropical Diseases.

Best regards,

Shaden Kamhawi

co-Editor-in-Chief

Paul Brindley

co-Editor-in-Chief
